# A New Method of Removing Fine Particulates Using an Electrostatic Force

**DOI:** 10.3390/ijerph18126199

**Published:** 2021-06-08

**Authors:** Jaeseok Heo, Yelim Jang, Michael Versoza, Gihwan Kim, Duckshin Park

**Affiliations:** 1Transportation Environmental Research Team, Korea Railroad Research Institute, Uiwang 16105, Korea; jsheo1005@krri.re.kr (J.H.); yelimm412@krri.re.kr (Y.J.); mikeverz23@krri.re.kr (M.V.); 2Railway System Engineering, University of Science and Technology, Daejeon 34113, Korea; 3Engineering Team, YK Eng., Yongin 17095, Korea; gihwan32@nate.com

**Keywords:** bus stops, shelters, electrostatic force, brush filter, fine particulates, rotation

## Abstract

Many studies have found that the concentration of fine particulates in the atmosphere has increased. In particular, when using the bus, the situation in which people are exposed to relatively high concentrations of fine particulates is increasing. The purpose of this study is to reduce exposure to these potentially harmful particulates by introducing open shelters at outdoor bus stops. In order to use it as an outdoor fine particulates reduction device, a brush filter using electrostatic force (EF) was used on an experimental scale and the generation of electrostatic force, according to the material, was examined. As electrostatic force was generated, the fine particulates collection performance was about 90% efficiency. In addition, it was confirmed that the efficiency of each particle size was improved by 57% through structural improvement. Finally, through experimentation, it was confirmed that the brush module can be used for about 70 days.

## 1. Introduction

Many studies have found that particulate matter (PM) is harmful to the human body [1,2,3,4,5]. Research shows that PM_2.5_ and PM_10_, in particular, are implicated in respiratory disease [6,7], asthma in children [8], and reduced lung function in teenagers [1,9].

In urban areas, vehicles are one of the major sources of PM emissions [3,10,11]. Internal combustion engines (ICE) in vehicles emit a variety of pollutants, such as PM, sulfur oxides, nitrogen oxides, and ozone [10]. The PM generated by ICE engines can lead to cardiovascular contraction, causing hypertension and heart disease [12,13]. Several studies have determined that high and continuous exposure to PM is associated with an increased relative risk of daily cardiovascular disease mortality [14,15,16]. When PM_10_ is increased up to 10 µg/m^3^, the cardiopulmonary mortality rate increases to 0.7% [17] according to NMMAPS (Nation Morbidity Mortality Air Pollution Study) data. In addition, Yin et al. (2017) reported that the effect of particulates on deaths from cardiopulmonary disease was 0.62% (0.43% to 0.81%) per 10 ug/m^3^ [18]. Moreover, Pope et al. (2006) found that a 10 µg/m^3^ increase in PM_2.5_ resulted in a 4.5% increase in acute ischemic coronary events [19]. In view of this previous research, it is evident that harm to the human body is reduced when PM_10_ is reduced through air quality improvement.

However, Li et al. (2009) and Zheng et al. (2015) reported that many people who use public transportation, namely buses, spend at least 10 min waiting at bus stops every day [20,21]. People can be exposed to high concentrations of pollutants while they are waiting for buses in city centers [22].

A study by Hess et al. (2010) found that people were exposed to PM while waiting inside and outside bus stops [23]. PM concentrations inside bus shelters with openings oriented toward the roadway were consistently higher than those measured outside the same bus stops [11,23]. Such findings demonstrate that people inside bus shelters are exposed to exhaust fumes and PM emitted from vehicles for a longer period than if they were waiting outside. Moore et al. (2012) found that the location and orientation of bus shelters had a significant influence on the degree of exposure of the occupants to PM and other pollutants, and reported that these facilities would be better able to function as shelters if their orientation were changed [11].

Existing PM removal technologies include electrification, artificial coagulation to convert PM into particles of a consistent size, and control of particle behavior or separation of particles [24]. In other words, technologies to remove airborne PM artificially apply external force or barriers to air flow to separate out the PM [24]. Regarding electric fine particulate removal methods, previous studies have mainly focused on generating an artificial electrostatic force with a corona discharge electrification system before and after operating a filter [25].

Some studies have found, however, that high charging voltages may generate harmful levels of ozone [26,27,28,29,30]. For this reason, this study created electrostatic force (EF) through friction, without high voltages, and sought to consistently generate that force using a motor. A motor makes protracted use of a brush filter possible, by generating a consistent electrostatic force through the contacts between the filter and friction plates. The brush filter was intended for outdoor use and is easy to replace and clean. At the experimental stage, its effectiveness was examined through its ability to remove PM, and its operating lifespan was tested in an indoor space.

This study is a preliminary experiment on the performance of a brush filter before installation at a bus stop in Korea. The aim of this study was to apply this technology in the actual field at a bus stop. The observations made in this study will also serve as a baseline for simulation comparison in future experiments.

## 2. Material and Methods

### 2.1. Electrostatic Force Test By Material and Rotation Speed

To generate frictional electricity, electrostatic force was created to reach the equilibrium of electric charge for each material [31]. Figure 1 orders various materials according to their electric charge. Following Ho and Cho (2018), PC and PVC friction plates were used to generate static electricity with a nylon brush filter [31].

For the experiment, friction plates were attached to both the upper and lower parts of a module, and a brush filter was set in the middle. As the motor rotated, the electrostatic force generated was measured through a static electricity sensor set at a distance of 1 inch from the brush filter. The experiment was conducted at filter rotation speeds of 30, 45, and 60 rpm to identify the effects of speed. Changes in electrostatic force were recorded after turning the brush filter on and off, to compare electrostatic force before and after filter rotation, and measured for 5 min each.

### 2.2. Removal of Fine Particulates by Rotation Speed

A PVC friction plate was used in this experiment to determine the filter’s fine particulates removal performance at each rotational speed. An A1 ultrafine test dust (A1 dust; Powder Technology Inc., Arden Hills MN, USA) was fed into a duct through a solid aerosol generator (SAG 410; TOPAS GmbH, Dresden, Germany), and PM concentrations were measured with a portable aerosol spectrometer (model 1.109; Grimm, Germany). During the experiment, the PM concentrations flowing into the brush filter were consistent (300 μg/m^3^ on average). Fine particulates removal was measured by checking the PM concentrations coming out of the brush module through the filter as the module rotated. This experiment measured the difference between the input and output PM concentration over time, for 60 min, according to the changing rotation speeds (0, 30, 45, and 60 rpm). The experiment was operated for 1 h.

The fine particulates removal efficiency (%) was calculated according to Formula (1):(1)$$Reduction\;Efficiency\;\left(\%\right)=\left(1-\frac{C_{pout}}{C_{pin}}\right)\times 100\;\;\;\;\;$$

In this formula, *C_pin_* refers to concentrations of particles flowing into the duct, while *C_pout_* refers to concentrations of particles coming out of the duct. This experiment examined the fine particulates removal performance of the brush module by using different brush filter rotation speeds (0, 30, 45, and 60 rpm).

### 2.3. Contact Time of Fine Particulates at the Brush Filter

Kim et al. (2018) found that fine particulates efficiency decreased as retention time decreased. Accordingly, structural improvements to increase retention improved the brush filter efficiency [32]. Figure 2 shows the structure before and after the improvement, which involved attaching a cover to the filter. The filter performance was tested before and after attaching the cover. As in the previous experiment, the input PM concentration was relatively the same, and the output concentration, with regards to the improved structure, was measured and compared with the results gathered before the improvement.

### 2.4. Prediction of Brush Filter Lifespan

While a high-efficiency particulate air (HEPA) filter provides better PM reduction performance than other filters, it is expensive and has high maintenance costs as it must be replaced regularly. Therefore, filters need to be chosen based on their price, the cost of maintenance, and their replacement cycle [33].

To determine the brush filter’s lifespan, an experiment was conducted under conditions of greater than measured atmospheric PM concentrations. Publicly available data from the Bugok 3-Dong measurement station in Uiwang, South Korea, was analyzed to determine atmospheric PM concentrations for use in the experiment. Table 1 shows the daily average atmospheric PM_10_ concentrations measured by the station.

In the experiment, to calculate the brush filter lifespan at concentrations of 300–350 μg/m^3^, which is 10 times higher than the average concentration of fine particulates in the atmosphere (~36 μg/m^3^), changes in the input and output were confirmed. Measurements were conducted over 15 days with the brush filter rotating at 45 rpm. The lifespan was estimated based on the change in efficiency of fine particulates removal during the experimental period and the replacement cycle was calculated.

## 3. Results and Discussion

The brush filter used in this study can be used for more than 2 months and is easy to replace. It can be reused, without disposal, after washing and so can be used semi-permanently. In addition, the brush filter showed excellent performance in reducing PM_2.5_, which plays a fatal role on human health. This device can reduce the concentration of ultrafine particulates. As mentioned by Pope et al. (2006), it can be inferred that reducing PM_2.5_ can reduce acute ischemic coronary events by 4.5%.

However, as mentioned above, this experiment does not include other factors such as the external environment, due to the fact that the experiment was conducted indoors.

To solve this problem, we plan to install a brush filter at a bus stop in the future to conduct measurements.

### 3.1. Effect of Material and Rotation Speed on EF

Figure 3 compares the results of generating electrostatic force between a brush module consisting of a nylon filter and PC friction plates, and a module consisting of a nylon filter and PVC friction plates at different rotation speeds over time.

As Figure 3 shows, during rotation of a brush filter, the PC friction plates generated electrostatic force of about (−) 2 kV, while the PVC friction plates generated about (+) 3 kV. This difference was due to the unique electric charges of these materials. PVC generated more electrostatic force than PC because of the greater difference in the amount of electric charge between PVC and nylon as compared to PC and nylon. Table 2 shows the average electrostatic force generated according to friction plate material and filter rotation speed.

### 3.2. Effect of Rotation Speed on Particulate Matter

To assess brush filter performance in terms of removing fine particulates, after inputting the test fine particulates at a consistent concentration of about 300 μg/m^3^, the output fine particulates concentration was measured at filter rotation speeds of 0, 30, 45 and 60 rpm. Figure 4 shows the results.

The filter efficiency was 74.0% at 0 rpm, which demonstrates the efficiency of the filter structure. The efficiency changed according to rotation speed, from 90.3 ± 1.9% at 30 rpm to 87.1 ± 2.1% at 45 rpm and, finally, back to 90.3 ± 2.0% at 60 rpm. These results suggest that electrostatic force is able to reduce PM and improve filter performance, regardless of rotation speed.

Table 3 shows the filter performance according to rotation speed.

The removal efficiency was compared among particle sizes, and an analysis was performed to determine the reason for fine particulates reductions.

Existing studies have found that electrostatic force generated by a filter can increase filtration efficiency without reducing pressure [25,34].

As stated above, this study found that electrostatic force improved the filter’s PM removal efficiency. In particular, in this experiment, electrostatic force removed a large amount of PM_2.5_. Figure 5 shows the removal efficiency of the brush filter for each particle size according to changes in the electrostatic force. In the figure, the filter was highly effective at removing dust particles of 4.884 μm or larger in size, regardless of the amount of electrostatic force, mainly due to the dominant influence of the brush module structure. For smaller particles, the electrostatic force had the largest influence on removal efficiency. From this result, it can be seen that, in the case of this brush filter, large PM_10_ removal depends more on the structure than PM_2.5_ removal, which can be achieved using electrostatic force.

### 3.3. Effects of Contact Time

The effects of structural changes in the brush filter on removal efficiency were analyzed by particle size class. Figure 6 shows the filter efficiency by particle size at 0 rpm, i.e., when no electrostatic force is generated. In the area where electrostatic force was dominant, particle removal efficiency improved by 57% on average after structural improvement. However, efficiency did not change when the brush module structure was the dominant factor in removal performance. It appears that the structural improvement had a greater effect on filter efficiency than electrostatic force, especially for removing particles smaller than 4.884 μm. Structural improvement increased the size range of particles removed by the filter, which is important with respect to PM removal from outdoor spaces.

### 3.4. Lifespan Prediction

Figure 7 shows the results of the experiment conducted to determine the brush filter’s lifespan. In the experiment, test dust was input at concentrations of 300–350 μg/m^3^, which are much higher than typical atmospheric PM concentrations. The filter showed a high filtration efficiency of about 88%. However, removal efficiency decreased as PM inflow increased over time (Table 4).

Over the course of the experiment, brush filter removal efficiency decreased from 88% to 71%. With a brush filter efficiency of 80%, the replacement cycle is 7 days. Thus, use for 70 days is possible based on the average daily atmospheric PM concentration.

Table 5 shows the PM_10_ holding capacity of the filter, which increased over time.

## 4. Conclusions

Previous studies found that people who use buses are exposed to high concentrations of pollutants, as they tend to spend at least 10 min at bus stops every day. According to one study, PM concentrations inside bus shelters with openings oriented toward the roadway were consistently higher than those outside the shelters. Thus, improvement of bus shelter design is necessary.

This study introduced an air-purifying system for outdoor use that can reduce PM exposure at, for example, bus shelters. The system uses a brush module that generates electrostatic force. Experiments were performed to determine the amount of static electricity generated by PC and PVC, fine particulates removal efficiency according to rotation speed, retention time, and filter lifespan.

PVC generated 3 kV of static electricity, which exceeds that generated by PC. The fine particulates removal efficiency was 90.3 ± 1.9%, 87.1 ± 2.1% and 90.3 ± 2.0% at filter rotation speeds of 30, 45, and 60 rpm, respectively.

The average fine particulates removal efficiency increased by 57% when the retention time was extended through structural improvement of the filter. In addition, based on the experimental results, it was concluded that the brush module could be used for about 70 days.

Based on these studies and results, it was found that the brush filter using electrostatic force is effective in reducing fine particulates.

Although this study was concerned with the outdoor use of a PM filter, the experiments were conducted indoors without consideration of performance changes that could be caused by environmental factors during outside use. Due to this limitation, more studies are needed on PM reduction efficiency, and other aspects of filter performance, when used outdoors. In the future, after improving the brush filter and installing it at a bus stop, experiments from both the laboratory and field will be measured and compared.

## Figures and Tables

**Figure 1 ijerph-18-06199-f001:**
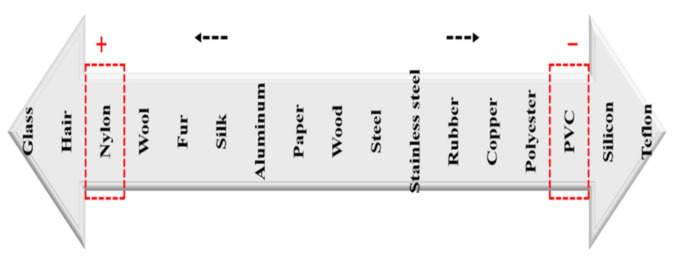
Electric charges of various materials.

**Figure 2 ijerph-18-06199-f002:**
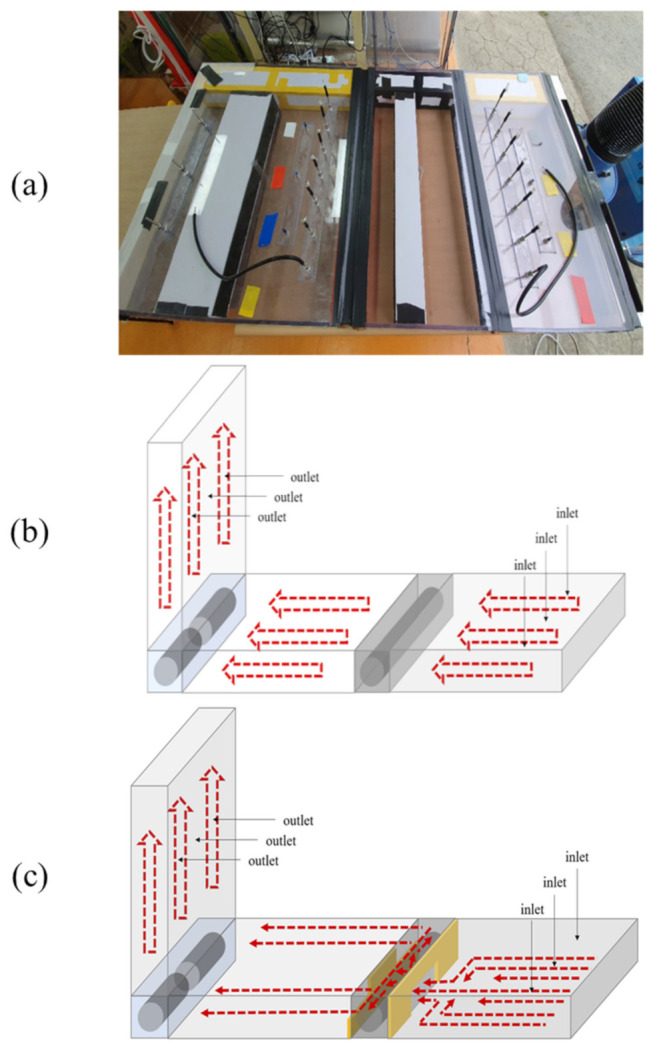
Setup of experiment to compare filter efficiency: (**a**) experiment setting photo, (**b**) before structural improvement, and (**c**) after structural improvement.

**Figure 3 ijerph-18-06199-f003:**
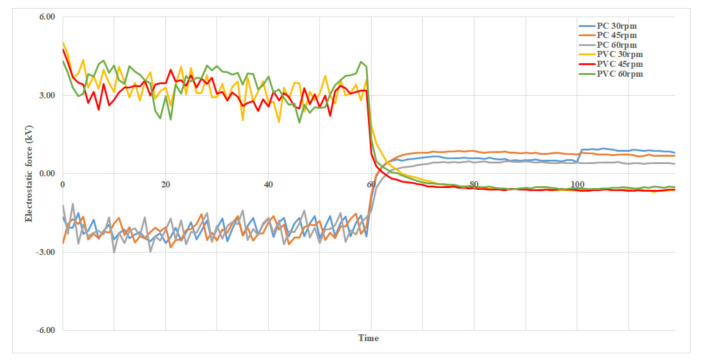
Electrostatic force generated according to friction plate material and filter rotation speed.

**Figure 4 ijerph-18-06199-f004:**
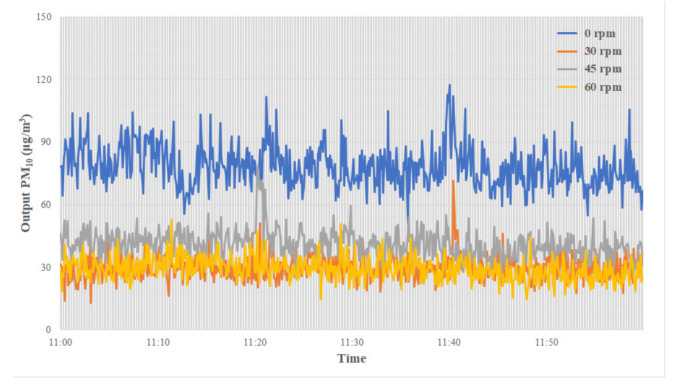
Changes in PM_10_ concentrations according to filter rotation speed.

**Figure 5 ijerph-18-06199-f005:**
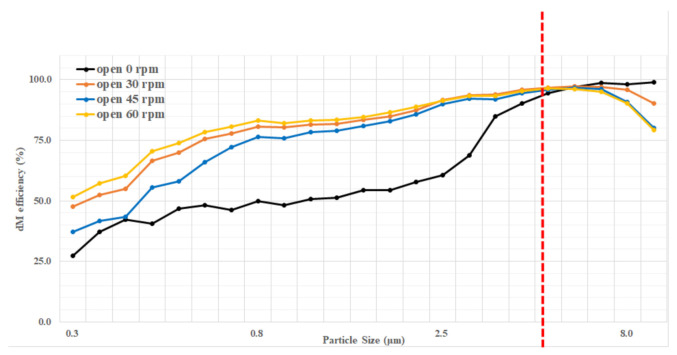
Changes in removal efficiency for each particle size.

**Figure 6 ijerph-18-06199-f006:**
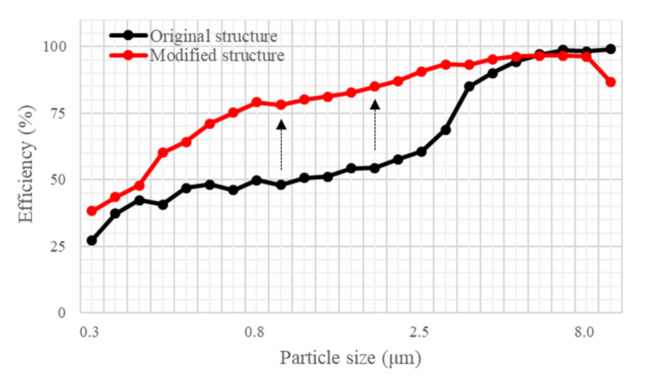
Particle size removal efficiency of the original and modified structure of the brush filter.

**Figure 7 ijerph-18-06199-f007:**
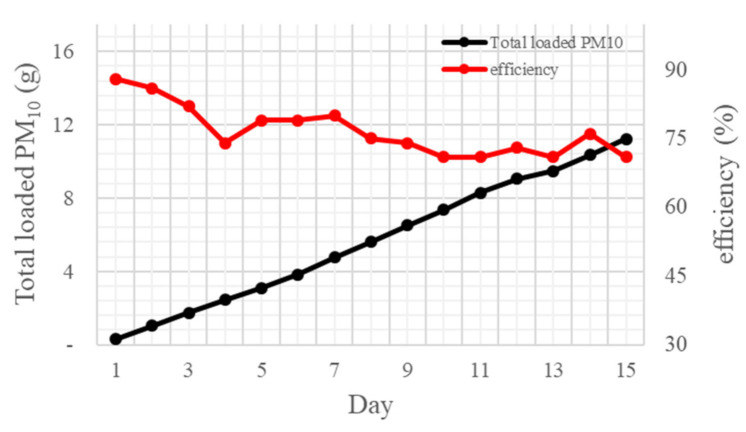
Filter efficiency according to the total PM amount.

**Table 1 ijerph-18-06199-t001:** Atmospheric PM concentrations during the experimental period in 2020 (Bugok 3-Dong measurement station).

Date	19 May	20 May	21 May	22 May	25 May	26 May	27 May	28 May
PM_10_ (μg/m^3^)	14	12	23	53	39	37	27	28
Date	29 May	1 June	2 June	3 June	4 June	5 June	8 June	average
PM_10_ (μg/m^3^)	36	20	26	57	54	65	50	36

**Table 2 ijerph-18-06199-t002:** Average electrostatic force (kV) generated by PC and PVC at different filter rotation speeds.

		30 rpm	45 rpm	60 rpm
PC	On	−2.11 ± 0.32	−2.16 ± 0.31	−2.13 ± 0.41
	Off	0.63 ± 0.28	0.72 ± 0.24	0.35 ± 0.28
PVC	On	3.30 ± 0.55	3.16 ± 0.46	3.44 ± 0.60
	Off	−0.39 ± 0.46	−0.51 ± 0.24	−0.40 ± 0.31

**Table 3 ijerph-18-06199-t003:** Average amount of fine particulates removed at each filter rotation speed.

	0 rpm	30 rpm	45 rpm	60 rpm
Input (μg/m^3^)	301.3 ± 35.2	308.2 ± 54.1	320.9 ± 56.1	303.2 ± 28.7
Output (μg/m^3^)	78.4 ± 10.0	30.0 ± 6.0	41.3 ± 6.7	29.5 ± 5.5
Efficiency (%)	74.0 ± 3.5	90.3 ± 1.9	87.1 ± 2.1	90.3 ± 2.0

**Table 4 ijerph-18-06199-t004:** Changes in PM_10_ removal efficiency over time.

Date	1	2	3	4	5	6	7	8
Efficiency (%)	88 ± 2.3	86 ± 3.0	82 ± 4.2	74 ± 3.9	79 ± 4.1	79 ± 17.2	80 ± 3.6	75 ± 4.4
Date	9	10	11	12	13	14	15	
Efficiency (%)	74 ± 6.1	71 ± 5.6	71 ± 5.8	73 ± 5.9	71 ± 6.0	76 ± 4.7	71 ± 5.7	

**Table 5 ijerph-18-06199-t005:** PM_10_ holding capacity of the filter over time.

No. of days	1	2	3	4	5	6	7	8	9	10	11	12	13	14	15
Holding capacity (g/h)	0.2	0.9	1.5	2.0	2.5	3.1	3.8	4.5	5.1	5.7	6.4	6.9	7.3	7.9	8.5

## Data Availability

Data sharing not applicable.

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
