# Peer review of "A New Method of Removing Fine Particulates Using an Electrostatic Force"

_ijerph, 2021, doi:10.3390/ijerph18126199_

Round 1

Reviewer 1 Report

The authors present a method for removing fine particulates present in air by a brush filter device using electrostatic force.

I think that the manuscript section about the description of the device, the tailoring of its features for achieving better 

results in removal and its durability are sufficiently good and clearly presented, except that when presenting the mean values for the parameters concerning the performance it is necessary to clarify the number of replicates done and the related uncertainity of the results.

My main concern is about the suitability for publication in IJERPH.

Even if in the introduction the attention is focused on PM ambient air pollution and on the effect that it can have on people waiting the bus in the bus shelter, the experiments presented are vary far from reproducing this condition, the authors stated that:

- par. 2.2 "In the experiment, the PM concentrations flowing into the brush filter were consistent (300 μg/m3 on average)."

- par. 2.4 "In the experiment to calculate brush filter lifespan, at 300–350 μg/m3, which is 10 times higher than the average concentration of fine particulates in the atmosphere (~36 μg/m3), changes in input and output were confirmed."

- Conclusions "Although this study was concerned with outdoor use of a PM filter, the experiments were conducted indoors, without consideration of performance changes that could be caused by environmental factors during outside use."

Thus, in my opinion, the study does not present the link between ambient (or indoor) air pollution and improvement of air quality conditions for human health.

I expected at least a test in a chamber with PM conditions similar to that in a bus shelter or, for example, in indoor air of a specific work environment. 

I think that the aforementioned experiments are necessary to link the use of the device to a benefit for human health.
Otherwise I think that this study is not suitable for publication in IJERPH.

Other observation:

- Introduction: I think that instead of "nitrates" (which are salts contained in PM), "nitrogen oxides" (which are gases) is more appropriate.

- A comparison with studies present in literature about the results of using similar devices for PM abatement in "real conditions" or "simulated real conditions" is necessary. 

- English checking is necessary for the abstract and some sentences at the end of the Introduction.

Author Response

Good Day!

Thank you for your time and effort. Attached is our response to your helpful and insightful comments. We hope we could here from you soon.

Respectfully,

Jaeseok Heo

Reviewer 2 Report

The manuscript  presents a certain concept of using the device to reduce  PM concentrations at bus stops, a brush filter using electrostatic  force.   The study   concerns the assessment of the effectiveness of this device for removing PM particles, taking into account two designs of the device, different rotational speeds of the brushes, and the change in efficiency over time. These issues are within the aims of the journal (Environmental science and engineering), but as it is an experimental work, the emphasis should be on the methodological part.  Please clearly state the subject and purpose of the research. Please attach a photo of the test stand and present it in the form of a diagram with the  installed measuring  devices.  Please describe in detail the research procedure, number of trials, duration of a single experiment, etc. The discussion   should include the advantages and disadvantages of the tested  device and a proposal to implement this solution, not general (as it is now), but detailed.

Please pay attention to the correct description of the axes in Figures 3.1 and 3.3  and that the conclusions  should refer to the main  test results and a proposal for future studies.

Author Response

(The authors gave the same response as above.)

Round 2

Reviewer 1 Report

The authors have revised the manuscript in some parts adding information, but I think that they have not answered to the main issue I rose.

It is not sufficient to add some sentences in the introduction to prove the link between the applicability of the device to outdoor/indoor conditions. This is theoretical.

I think that more EXPERIMENTS are necessary. At least some experiments that show the PM abatement for a PM concentration range similar to outdoor or indoor conditions. For example 40-60 ug/m3 for outdoor and 80-120 ug/m3 for indoor (some work environments).

Moreover in the result tables the number of replicates and standard deviations have to be absolutely included.

Author Response

(The authors gave the same response as above.)

Reviewer 2 Report

I believe that the readability of the manuscript  has improved enough to be published.

Author Response

Good Day!

Thank you for your time and effort.

Respectfully,

Jaeseok Heo

Round 3

Reviewer 1 Report

The authors answered to my comments, I think that the manuscript has been improved.